# An Insight into Occurrence, Biology, and Pathogenesis of Rice Root-Knot Nematode *Meloidogyne graminicola*

**DOI:** 10.3390/biology12070987

**Published:** 2023-07-11

**Authors:** Arunachalam Arun, Annaiyan Shanthi, Muthurajan Raveendran, Nagachandrabose Seenivasan, Ramamoorthy Pushpam, Ganeshan Shandeep

**Affiliations:** 1Department of Nematology, Tamil Nadu Agricultural University, Coimbatore 641003, Tamil Nadu, India; 2Directorate of Research, Tamil Nadu Agricultural University, Coimbatore 641003, Tamil Nadu, India; 3Department of Rice, Tamil Nadu Agricultural University, Coimbatore 641003, Tamil Nadu, India

**Keywords:** *Meloidogyne graminicola*, rice root-knot nematode, morphometrics, diagnosis, ITS rDNA, molecular docking

## Abstract

**Simple Summary:**

Rice root-knot nematode is a plant-parasitic nematode that infects the roots of rice plants. It is also known as *Meloidogyne graminicola*, and is one of the most damaging pests of rice crops worldwide. The nematode infects the roots of rice plants and causes the formation of characteristic knots or galls. Both morphological and molecular characterization can be used in combination to provide a more complete understanding of nematodes. Molecular analysis can be used to identify new nematode species, while morphological analysis can be used to describe their physical features and provide a more complete picture of their biology and ecology. The current study aimed at utilizing both morphological and molecular characterization, and life stage as well, as molecular aspects of interaction between the rice root-knot nematode *Meloidogyne graminicola* and its host plants.

**Abstract:**

Rice (*Oryza sativa* L.) is one of the most widely grown crops in the world, and is a staple food for more than half of the global total population. Root-knot nematodes (RKNs), *Meloidogyne* spp., and especially *M. graminicola*, seem to be significant rice pests, which makes them the most economically important plant-parasitic nematode in this crop. RKNs develop a feeding site in galls by causing host cells to differentiate into hypertrophied, multinucleate, metabolically active cells known as giant cells. This grazing framework gives the nematode a constant food source, permitting it to develop into a fecund female and complete its life cycle inside the host root. *M. graminicola* effector proteins involved in nematode parasitism, including pioneer genes, were functionally characterized in earlier studies. Molecular modelling and docking studies were performed on *Meloidogyne graminicola* protein targets, such as β-1,4-endoglucanase, pectate lyase, phospholipase B-like protein, and G protein-coupled receptor kinase, to understand the binding affinity of Beta-D-Galacturonic Acid, 2,6,10,15,19,23-hexamethyltetracosane, (2S)-2-amino-3-phenylpropanoic acid, and 4-O-Beta-D-Galactopyranosyl-Alpha-D-Glucopyranose against ligand molecules of rice. This study discovered important molecular aspects of plant–nematode interaction and candidate effector proteins that were regulated by *M. graminicola*-infected rice plants. To the best of our knowledge, this is the first study to describe *M. graminicola*’s molecular adaptation to host parasitism.

## 1. Introduction

Rice (*Oryza sativa* L.) is among India’s foremost staple food crops, providing calories to over sixty percent of the country’s population, and influencing the livelihoods and economics of several billion people, primarily in Asia, Latin America, the Middle East, and the West Indies. Rice has influenced Asian societies and cultures for ages. Many Asian societies have a relationship with rice that goes beyond the fulfilment of fundamental requirements [1]. There are always new threats to plant health. *Meloidogyne graminicola*, a nematode that causes rice root-knots and is suited to flooded environments, is one example of this. It poses a threat to all varieties of rice agrosystems. It was recently discovered in Italy, and has been added to the EPPO alert list (European and Mediterranean Plant Protection Organization) [2]. RKNs are capable of penetrating the roots, causing root galling, suppressing plant defence mechanisms, hijacking the plant’s metabolism, and establishing giant cells for their benefit [3]. As a result, plants start losing vigour, resulting in significant productivity losses [4]. *Meloidogyne graminicola* has evolved to be the most problematic rice parasite among some of the several RKNs [5,6]. Upland rice (13%) is rainfed, although there is no surface water accumulation. It is impacted by a variety of biotic and abiotic stressors, with plant-parasitic nematodes playing a significant role in these pressures. In the constantly evolving rice production system, there are over 200 species of phytonematodes that have been recognized to be related to rice [7]. *M. graminicola* is a major rice-parasitic nematode that represents a significant danger to rice farming, especially in Southeast Asia, where almost 90% of the global rice is cultivated and consumed. Terminal, hook-shaped, or spiral-tip galls are typical signs of *M. graminicola* infestation, and are produced by this nematode species [5]. The rice root-knot nematode has become a pest with global significance. Under the simulated conditions of intermittently flooded rice conditions, *M. graminicola* causes yield losses ranging from 11 to 73%, while under simulated upland conditions, loss of yield ranged from 20 to 98% [8]. The only way to effectively control *Meloidogyne* species is to quickly and precisely identify the nematode. Conventional means of nematode diagnosis, based on morphometric characteristics, are difficult since they are time-consuming and call for substantial training and experience [9]. Earlier, phytonematodes were diagnosed solely through morphometric traits. At present, less information is available in India about the molecular characterization of *Meloidogyne* spp., particularly *M. graminicola*. *Meloidogyne* species can be identified using a variety of molecular approaches [10]. Polymerase chain reaction (PCR) is a sensitive, rapid, and precise method among all these strategies [11], being a molecular detection key that employs numerous molecular techniques to detect seven economically significant RKNs commonly seen in diagnostic laboratories. Furthermore, with the rise in DNA-based sequencing, the tandem repeat unit segments of the 18S, ITS1, 5.8S, ITS2, and 28S regions of the ribosomal DNA array (rDNA) and mitochondrial DNA (mtDNA) have proven to be efficacious diagnostics for characterizing RKNs [12]. For field management techniques to be effective, rapid diagnosis of *M. graminicola*, as well as knowledge of its incidence and dissemination range, is essential. Therefore, to ascertain the spread, severity, and severity of the disease caused by *M. graminicola* in Tamil Nadu, India, we used morphometrics and molecular techniques to identify RKNs. The effector proteins from dorsal esophageal glands of *M. graminicola*, such as pectate lyase [13], phospholipase B-like protein [14], and G protein-coupled receptor kinase [15,16] favour parasitism with rice plants. Hence, we attempted to elucidate the mode of pathogenicity action of these molecules towards the rice plants. However, the details of ligand binding with nematode target proteins have not been investigated in order to discover interactions between the *M. graminicola* and the host plant. Understanding how small molecules or ligands interact with the nematode target sites will aid in elucidating the mechanism of parasitic behaviour.

## 2. Materials and Methods

### 2.1. Survey and Sampling

A routine investigation for the existence of root-knot nematode infestations was made in rice fields in Tamil Nadu, India throughout 2021 and 2022 (Table 1). Based on the crops’ aboveground abnormalities, such as wilting, stunted growth, yellowing of the leaves, and root galls, nematode-infested fields were detected (Figure 1). At a depth of 10–15 cm, midseason soil and root samples were taken from the paddy rhizospheres.

Five plants were randomly chosen from each locality, and three samples were taken from each plant. The soil samples were carefully combined, and a 200 cc composite sample was taken in polythene bags and labelled appropriately for examination. Additionally, galled or infested roots were gathered (Figure 2, Table 2). The samples were transported to the lab for nematode extraction while being maintained in plastic bags in an ice box.

### 2.2. Morphological Characterization of Meloidogyne spp. Infesting Rice

Nematodes were extracted by using Cobb’s sieving and decanting method with a modified Baermann’s funnel method. Then, they were killed in hot formalin at 65 °C and fixed in a formalin:acetic acid fixative (FA 4:1) [17]. The nematode specimens were dehydrated through the rapid glycerin (Seinhorst’s) method and mounted in pure glycerin on glass slides supported by glass rods of a diameter slightly larger than that of the nematodes [17]. Adult females were extracted, cleaned of dirt from tip-galled roots, and thoroughly separated from galls, from which posterior cuticular patterns (PCPs) were acquired for species diagnosis. A Nikon eclipse Ti2-U inverted microscope was used for obtaining microphotographs of posterior cuticular patterns. The morphometrics were analyzed by using NIS-Elements Denoise.ai software. Galled roots were also diagnosed for the presence of any egg mass. The rice root-knot nematode was examined using the criteria of Ye and Hunt [18] for species diagnosis.

### 2.3. Molecular Characterization of Meloidogyne graminicola Infesting Rice

DNA samples were prepared according to Castagnone-Sereno et al. (1995) [19]. In the PCR investigations, three sets of primers (synthesised by Eurofins, Bangalore, India) were employed to amplify the ITS D2-D3 expansion portions of 28S rRNA, as well as the coxII region. ITS-F (5′-GTT TCC GTA GGT GGT GAA CCT GC-3′) and IT-S R (5′-ATA TGC TTA AGT TCA GCG GGT-3′) primers were used to amplify the ITS [20]. The forward D2A (5′-ACA AGT ACC GTG AGG GAA AGT TG-3′) and reverse D3B (5′-TCG GAA GGA ACC TAC TA-3′) primers were used for amplification of the D2-D3 28S rRNA gene [21]. The mitochondrial region between the partial coxII and the partial 16S was amplified using the forward primer COI-F (5′-TTT TTT GGG CAT CCT GAG-3′) and the reverse primer COI-R (5′-AGC ACC TAA CTT AAA C-3′) [22]. The PCR conditions were followed as specified by Ye and Hunt [18].

All of the PCR amplifications were performed separately in 25 µL of reaction volumes containing 2.0 µL of DNA, 1.0 µL of each 10 µM primer (forward and reverse), 2.5 µL of 10X buffer, 1.5 µL of 200 mM of each dNTP, and 2 units of Taq polymerase enzyme, and the final volume was prepared with approximately 25 µL MilliQ nuclease-free water. Amplified PCR products were sequenced by the Sanger dideoxy method (Eurofins, Bengaluru, India). The derived sequences were blasted with NCBI BLASTN for phylogenetic analysis. The phylogenetic tree was constructed by using MEGA7 software. An appropriate model, as established by MEGA7, was utilized to generate the evolutionary origins using maximum likelihood analysis. One thousand replications were used to construct the bootstrap consensus tree. Branches associated with partitioning that were only replicated in 70% or fewer bootstrap replicates were collapsed. Initial trees for the heuristic search were automatically generated using the neighbor-joining and BioNJ algorithms on a pairwise distance matrix calculated using the JTT model, and the topology with the highest log likelihood value. Regions with gaps and incomplete data were removed.

### 2.4. Life Cycle Analysis

Under glasshouse conditions, the nematode culture was kept in potted plants of the rice variety TN 1. The aforementioned kind of seed was immersed in tap water for 24 to 48 h. The sprouting seeds were planted on 18 October 2022, in soil that was taken from a paddy field that had been infected with *M. graminicola*, and contained one second-stage juvenile (J2) per gram of soil. On 23 October 2022, a nutrient solution containing N, P, K, and Zn was added. Three plants were involved, and they were periodically uprooted so that life cycle research could be conducted. Under a stereo binocular microscope, the roots were stained with 0.1% acid fuchsin lactophenol to reveal the nematode’s various developmental phases [23]. A binocular microscope was used to observe the nematode at various stages of development.

### 2.5. Histopathological Studies of Roots Infested by M. graminicola

Root samples from rice plants that had been severely infested with root galls caused by the root-knot nematode *M. graminicola* were used in this study. Roots were cleaned to remove any remaining soil, cut into 0.3 cm pieces with a sharp blade, fixed in FAA (formalin acetic acid ethanol) for 36 h, dehydrated using a graded ethanol–xylol series, embedded in paraffin wax (melting point: 56–58 °C), and sectioned at a thickness of 8 µm using a hand rotary microtome. The sections were mounted in DPX mountant, and the sections were stained with safranin and counterstained with fast green [23]. The sectioned tissues were visualised using a Nikon eclipse Ti2-U inverted microscope.

### 2.6. Interaction between M. graminicola Effector Proteins and the Rice Plants

To understand the pathogenicity pattern of *Meloidogyne graminicola* parasitic proteins against rice cultivars, molecular modelling and docking studies were performed on protein targets such as pectate lyase [13], phospholipase B-like protein [14], G protein-coupled receptor kinase [15], and β-1,4-endoglucanase [16].

#### 2.6.1. Protein Target Identification and Molecular Modelling

Based on a review of the literature, the proteins pectate lyase, phospholipase B-like protein, G protein-coupled receptor kinase, and β-1,4-endoglucanase were identified as potential targets of the rice root-knot nematode *M. graminicola*. Based on literature mining, the protein sequences of selected targets for the rice root-knot nematode *M. graminicola* were retrieved from the UniProt database. The rice root-knot nematode’s chosen virulent target, *M. graminicola*, lacks experimentally and computationally solved structures. Therefore, SWISS-MODEL (method: rigid-body assembly), Phyre2 (method: profile-based alignment), and ROBETTA were used for molecular modelling (Metaserver). First, a model of each target sequence was created using the SWISS-MODEL server. Based on query coverage performance, SWISS-MODEL and Phyre2/ROBETTA were used for structure modelling in the absence of templates. The homology modelling for the target’s pectate lyase, phospholipase B-like protein, and G protein-coupled receptor kinase was created using the software SWISS-MODEL. Phyre2 was additionally used to model β-1,4-endoglucanase. Pectate lyase, phospholipase B-like protein, G protein-coupled receptor kinase, and model β-1,4-endoglucanase were among the protein targets in the homology modelling protocol that underwent BLAST search followed by HHblitsin SUZUKI-MODEL [24]. Maximum query coverage and a global mean quality estimation (GMQE) score of close to 1 were used as the parameters to ensure the high quality of modelled structures (Appendix A). Using comparative modelling domains, the ROBETTA server http://robetta.bakerlab.org/ (accessed on 11 April 2022) models multichain complexes using RoseTTAFold.

#### 2.6.2. Testing of the Protein Model

To assure model quality based on the residues residing in preferred and permitted regions, modelled protein targets were checked using the Ramachandran plot of the PROCHECK tool from the structural analysis and verification service (SAVES, Meta server) https://saves.mbi.ucla.edu/ (accessed on 18 April 2022). The Swiss PDB viewer was used to generate loops for residues in forbidden regions, and to minimise energy in proteins that were modelled “http://www.expasy.org/spdbv/” (accessed on 20 April 2022).

#### 2.6.3. Preparation and Analysis of Ligands

Three substances, Beta-D-Galacturonic Acid, 2,6,10,15,19,23-hexamethyltetracosane, and (2S)-2-amino-3-phenylpropanoic acid were gathered from the Pubchem database. The Pubchem database was accessed in SDF format to obtain 4-O-Beta-D-Galactopyranosyl-Alpha-D-Glucopyranose. Compounds were converted from the SDF file format to the PDB file format using the programme Open Babel v 2.3.1.

#### 2.6.4. Small Molecule Similarity Analysis

The similarity score between the compound pairs was calculated using the ChemMine online software. Tanimoto coefficients such as atom pair Tanimoto (AP), maximum common substructure Tanimoto (MCS), MCS size, MCS min/max, and SMILES were used to investigate the structural similarity of small molecules. Tanimoto is defined as c/(a + b + c), where c represents the number of features in a compound pair, a represents features unique to one compound, and b represents features unique to another. For atom pairs, the Tanimoto coefficient ranges from 0 to 1, with a larger coefficient indicating greater similarity (similar structural descriptors). Compounds with large Tanimoto differences (0.20) and the MCS make obtaining the most accurate and sensitive similarity measure possible. Since it is likely that similar molecules will share a large MCS size (9), the similarity analysis was established.

#### 2.6.5. Molecular Docking and Virtual Screening

PyRx 0.8’s AutoDock vina module was used to perform molecular docking [9]. The make macromolecule option in PyRx software was used to prepare the protein. The conjugate gradient, first-order derivatives of an optimization process with 200 steps, and commercial molecular mechanics parameters unified force field were used to minimise all ligand structures (UFF). To find binding site pockets for the targets, the Computed Atlas Topography of Proteins CASTp 3.0 server was used [25]. AutoDock4 and autogrid4 parameter files were used to set the grid and dock. During the docking protocol execution, ligands were allowed to generate flexible conformations and orientations with a value of 8 exhaustiveness. BIOVIA Discovery studio client 2021 https://www.3ds.com/products-services/biovia/ (accessed on 23 May 2022) was used to visualise interactions of docked conformations of protein–ligand complexes. To distinguish the receptor, ligand, and interacting atoms, different colours were assigned to them.

## 3. Results

### 3.1. Incidence and Dispersion of Meloidogyne graminicola in Tamil Nadu

Among the major rice-growing regions of Tamil Nadu, India surveyed for the occurrence of rice root-knot nematode *M. graminicola*, Coimbatore, Villupuram, Ariyalur, and Krishnigiri districts of Tamil Nadu were encountered with nematode infestation symptoms. Patches in the crop canopy and poor plant growth, including a stunted look and chlorotic leaves, were *M. graminicola*’s aboveground symptoms. Belowground symptoms were swollen and hooked root tip gall ends on infested root systems (Figure 2). Galled roots were examined, and it was discovered that they included both males and females. Pear-shaped females were revealed entrenched in the cortical layer of the galled root. The gall index [9] for rice root-knot nematode ranged from 3 to 4 where the districts encountered root-knot nematode symptoms (Table 3).

### 3.2. Morphometrics

**Posterior cuticular pattern (PCP)**: the female’s perineal patterns were oblong-shaped, with fine striae, dorsal semicircular arches, occasionally very few lines converged at each end of the vulva, and unclear or nonexistent lateral fields (Figure 3). Since 1949, morphological diagnosis based on the perineal pattern is one of the traditional methods for classifying *Meloidogyne* species [9]. The perineal pattern of isolates from Tamil Nadu, as determined by light as well as phase contrast microscopy, are somewhat different from previously reported patterns of *M. graminicola* [26,27], and overlap with both the patterns of *M. trifoliophila* and *M. oryzae*.

**Female:** the body is pearly white, pear-shaped, and has a short neck. The cuticle of the body has been annulated. The head is not clearly separated from the neck. The cephalic framework is weakly sclerotized. Small and delicate stylet with rounded knobs sloping backward. The oesophagus is substantially developed, with an elongate cylindrical procorpus and a broad, rounded metacorpus with a strongly sclerotized valve. Dorsal oesophageal gland aperture is located 3.9 µ (3.6–5.9) posterior to the base of the stylet (Table 3). Excretory pore distinct, one and a half stylet lengths from the stylet’s base. Two prodelphic convoluted ovaries. Vulva and anus are both terminally placed.

**Male**: body cylindroid, vermiform, tapering gradually at both ends. Body width: 34.6 µ (26.5–34.7). The head and body are not clearly separated. The cephalic framework is prominent. Cuticular annulation is extremely apparent. Blunt lips with a pronounced annulus and a longer spear conus than the shaft. Knobs are obvious and prominent. Excretory pore situated towards the back of the isthmus. Four lines form a lateral field. Stylet stout with rounded knobs. Dorsal oesophageal gland orifice 3.30 µ (2.8–4.0) posterior to the stylet base. Metacorpus elongates with a well-developed sclerotized valve. Spicules are arcuate, 25.6 µ (19.8–30.1) long. Tail length 12.1 µ (9.8–15.1), phasmids small, postanal situated around midway down the tail (Table 4, Figure 4G).

**Juvenile**: the body is cylindrical and vermiform, tapering towards the posterior body. The average body width is 16.9 µ (13.8–2.9). The head is not offset from the body. Cuticular annulations are small and conspicuous. Small and delicate stylet with rounded knobs sloping posteriorly. Aperture of the dorsal oesophageal gland 3.2 µ (2.8–3.7) posterior to the stylet’s base. Hemzonid, corresponding to three body annuli in length, just anterior to excretory pore. Spherical median bulb with pronounced sclerotized valve. Tail 71.4 µ (67.5–73.2) in length. The hyaline section of the tail terminal is 18.2 µ (16.3–19.2) in length and lacks a regular and conspicuous annulation. Tail end that is rounded and slightly clavate (Table 5, Figure 4A–F and Figure 5).

### 3.3. Histopathology

The presence of adult females and eggs was observed inside the cortical region of the root galls of susceptible rice plants with heavily infested galls, while groups of abnormally enlarged cells (giant cells) were common in the stele region (Figure 6). Eggs were laid in a gelatinous matrix, with egg masses embedded in the cortical region. Eggs were freed from the old roots by breaking the epidermal layer. In young roots, eggs hatch within the root. The juveniles or immature females persisted in the maternal gall or migrated intercellularly through the aerenchymatous tissues of the cortex to other feeding sides within the same root. Cell division and hypertrophy occurred as a result of larval inter- and intracellular movement in the root cortex (Figure 6). Juveniles gained access to the protophloem sites by destroying the pericycle cells. At larval establishment locations in the stele, aberrant xylem grew surrounding the large cell, producing expansion of the vascular cylinder. The histopathological analysis of nematode-infected rice roots revealed the development of “giant cells”, or specialized feeding sites, which were modified procambial cells in the stele region of the roots. Multinucleated giant cells with large vacuoles and dense cytoplasm were visible in the section. The infected root tissues had severe dislocation of xylem and phloem vessels. The number of giant cells varied from eight to nine. Around the giant cells, hollow cavities limited to the cortex were formed. The presence of three to four groups of young and old giant cells was shown in cross sections of galled roots (Figure 6). The newly derived giant cells were multinucleate and cytoplasmic. Older giant cells were vacuolated and devoid of cytoplasm. Cell wall thickening was noticed around the giant cells. The formation of giant cells clogged xylem arteries and sieve tubes. Cortical cell hypertrophy and hyperplasia were observed, which played a role in the development of root galls.

### 3.4. In Silico Analysis of M. graminicola Effector Proteins

Pectate lyase, one of the rice root-knot nematode parasitic proteins, was modelled using SWISS-MODEL utilising a template protein with 46.59 percent identity, 91 percent coverage, and a 0.72 GMQE score that had previously been described by electron microscopy (PDB ID-1EGZ). When experimentally solved using the X-ray crystallography method, the template for the phospholipase B-like protein (PDB ID-KAF7) had a GMQE score of 0.78, 97.18 percent identity, and 79 percent coverage. Since there were no homologs or templates for another nematode parasitic protein target β-1,4-endoglucanase in the SWISS-MODEL database, Phyre2 (http://www.sbg.bio.ic.ac.uk/phyre2/) (accessed on 12 May 2022) was used. Using the UniProt ID A0A8S9ZFJ9, the target protein sequence for β-1,4-endoglucanase (467 residues) was found. The β-1,4-endoglucanase protein model had a 100% confidence score and 87 percent coverage, and was based on protein structures such as PDB ID-6GJF. A G protein-coupled receptor target was modelled using I-TASSER with a 60% confidence score.

#### 3.4.1. Validation of Models

According to the Ramachandran plot, the target, phospholipase B-like protein, had 85.7 percent of its residues in the most favoured region or core region, 11.7 percent in additionally allowed regions, and 1.3 percent in generously allowed regions (Appendix A). Residues in the G protein-coupled receptor targets allowed, additionally allowed, and generously allowed regions are 81.9 percent, 13.3 percent, and 2.9 percent, respectively (Appendix A). The target β-1,4-endoglucanase had 75.2 percent of its residues in the most favoured region, 19.5 percent in additional allowed regions, and 3.5 percent in generously allowed regions (Appendix A). The target pectate lyase contained 76.7 percent of its residues in the most favoured region, 17.8 percent of its residues in additional allowed regions, and 3.9 percent of its residues in the generously allowed region (Appendix A).

#### 3.4.2. Analysis of Sequence Similarity

To determine the presence of any similar proteins in rice plants, sequence similarity was performed using the BLASTP tool for the nematode target proteins as a query against the rice genome proteins. In the similarity search, no single hit or similar sequences were found. Thus, specific binding of Beta-D-Galacturonic Acid, 2,6,10,15,19,23-hexamethyltetracosane, 4-O-Beta-D-Galactopyranosyl-Alpha-D-Glucopyranose, and (2S)-2-amino-3-phenylpropanoic acid to nematode protein targets was observed, but no binding to rice proteins was observed.

#### 3.4.3. Molecular Docking and Virtual Screening


**Beta-D-Galacturonic Acid**


Molecular docking and virtual screening modelled protein structures were docked with various compounds to determine their mode of binding (Figure 7 and Figure 8, Table 6). Beta-D-Galacturonic Acid had the binding affinity value of −5.1 kcal/mol with the target β-1,4-endoglucanase (H-bonds: THR 354, ALA 356, THR 359, THR 361), −5.8 kcal/mol (H-bonds: LYS 237, LYS 271) for G protein-coupled receptor, and binding affinity of −5.4 kcal/mol (H-bonds: GLU 154, LEU 162) for phospholipase B-like protein (Figure 8, Table 6). All complexes contained hydrogen bonds, indicating the stability and binding strength of Beta-D-Galacturonic acid to *M. graminicola* parasitic protein targets.


**2,6,10,15,19,23-hexamethyltetracosane**


Compound 2,6,10,15,19,23-hexamethyltetracosane had a binding affinity of −6.0 kcal/mol with target β-1,4-endoglucanase (H-bonds: THR 332, THR 334, THR 335, THR 392), −7.3 kcal/mol for G protein-coupled receptor protein target of *M. graminicola* (H-bonds: LYS 277), −7.7 kcal/mol for phospholipase B-like protein (H-bonds: TYR 133 and GLU131), and −7.5 kcal/mol for pectate lyase (H-bonds: TYR133, GLU131) (Figure 8, Table 6).


**4-O-Beta-D-Galactopyranosyl-Alpha-D-Glucopyranose**


The binding affinity value of 4-O-Beta-D-Galactopyranosyl-Alpha-D-Glucopyranose was −5.2 kcal/mol (H-bonds: GLU 156, LYS 283) with the target β-1,4-endoglucanase, −7.2 kcal/mol (H-bonds: LYS 237, THR 273) for G protein-coupled receptor, −5.5 kcal/mol (H-bonds: GLN 313) for phospholipase, and −6.9 kcal/mol (H-bonds: PRO 52) for pectate lyase. For each target, the H-bond donor and acceptor groups have been represented by a 3D docked complex (Appendix A).


**(2S)-2-amino-3-phenylpropanoic acid**


The binding affinity value of (2S)-2-amino-3-phenylpropanoic acid was −5.2 kcal/mol (H-bonds: ILE 75, TYR 149) with the target β-1,4-endoglucanase, −6.7 kcal/mol (H-bonds: ASP 279, ALA 305, SER 318, ASN 306 and ASP 319) for G protein-coupled receptor, −5.5 kcal/mol (H-bonds: ARG 72) for phospholipase, and −6.4 kcal/mol (H-bonds: VAL 53) for pectate lyase. In docked complexes, hydrogen bonds are formed with the backbone and side-chain of binding site residues (Figure 7, Table 6).

## 4. Discussion

The general morphology and morphometrics of the *M. graminicola* population in Tamil Nadu, India is in satisfactory correlation with the original descriptions [5]. According to the results of the current study, *M. graminicola* females were compared to previous morphometric findings [7]. Adult female populations were slightly varied in body length (474–594 µm) and DOGO value (3.6–5.9 µm). The current findings show that the entire length (1096.1–1731.4 µm) of the male *M. graminicola* was identical to Golden and Birchfield’s original description (1965) [5]. The highest body width (16.9 µm) of second-stage juveniles was encountered from Tamil Nadu populations when compared to earliest morphometric studies (Table 4). The sequence size of ITS, coxII-16S rRNA, and D2-D3 region of 28S are 399, 790, and 764 bp, respectively. All of the amplified sequences were comparable, and shared 99–100% identity with NCBI GenBank sequences for *M. graminicola*. For the *M. graminicola* Tamil Nadu population, novel sequences were discovered and deposited in NCBI GenBank with the accession codes OP712502 for ITS, OP714360 for D2-D3 of 28S, and OP714470 for coxII-16S rRNA genes. Based on ITS-rDNA, D2-D3 of 28S, and coxII-16S rRNA sequences, molecular identification of Tamil Nadu populations of *M. graminicola* was performed in the current study. In contrast with using molecular diagnosis, it is thought that variations in morphological traits, such as the stylet length, spicule, and vulva, are important for diagnosing the presence rice root-knot nematode *M. graminicola*. Our findings concur with the findings of [17], showing that molecular studies are more trustworthy for identifying *M. graminicola* since minor morphometrical variances and observed morphological differences may be related to regional distribution. The entry of *M. graminicola* second-stage juveniles (J2) began on the fourth day after sprouting rice seeds. Juveniles (J2) can enter the rice root tip regions at 48–72 h after hatching, and their number in the roots increases with time. Invaded juveniles establish their feeding sites in vascular tissues and reach the J3 stage after 120–144 h of intensive feeding. After 72–96 h of the J3 stage, the J4 stage starts, and the Casper adult stage is formed after another 96–120 h. A fully developed adult stage begins 264–312 h into the preadult stage. The roots had records of all larval stages, as well as adult males and females. Thus, until the 14th day after seeding, the second-stage juveniles in the soil kept attacking the roots. There were no second-stage juveniles found after the 15th day, and only adult stages were discovered after the 16th day (Figure 9). The histopathological studies revealed that rice root-knot nematode *Meloidogyne graminicola*-infested roots showed enlargements of giant cells with multiple nuclei at the stele region of the vascular system, and almost all of the females were mature, with a few being associated with egg masses. On the twentieth day, the maximum number of adults per root system (15–20 females and 2–3 males) was noted. Egg laying was noticed on day 20. Eggs were deposited both inside and outside of the gall system and wrapped in a viscous matrix. Six to seven members of the second-stage juveniles were encountered on the 24th day after sowing. *M. graminicola*’s life span on rice lasted 26–28 days to complete. The current findings support those of [31], who observed egg-laying females on days 20 to 24 following *M. graminicola* inoculation on paddy seedlings. The authors of [31] showed that this nematode completes its life cycle in 26–51 days on paddys throughout the year in eastern areas of India, highlighting the influence of temperature on life cycle duration. The presence of a protractible stylet, which serves to withdraw nutrients from the giant cells, as well as to release molecules, including virulence effectors, into the apoplast or directly into the host cells, is a distinguishing feature of plant-parasitic nematodes. Plant-parasitic nematodes secrete effector molecules into plant tissues to facilitate infection, reconfigure the cellular metabolism, or dissuade the accomplishment of plant defence responses.

Sedentary nematodes primarily release effectors produced in their esophageal glands into host tissues via their stylet, though other secretory organs may also play a role in the parasitic process [32,33]. Nematode management in large scale commercial rice cultivation was based on two to four nematicide uses, which are extremely harmful to soil microbial populations and biodiversity. It is, however, only a temporary solution because the nematode population grows vastly disproportionate after a few months, necessitating frequent applications of nematicides, which inevitably become toxic to the environment and uneconomical. Because of the negative effects of nematicides, scientists are eager to develop an alternative nematode management strategy. Virtual screening methods such as molecular docking had a significant impact in identifying a promising novel target site for the management of *M. graminicola*. We investigated potential protein target sites with rice biomolecules to use molecular docking to detect a molecule with the highest binding affinity on different *M. graminicola* target sites. The maximum binding (−7.7 kcal/ mol) of phospholipase B-like protein with the ligand 2,6,10,15,19,23-hexamethyltetracosane could promote nematode parasitic activity by regulating lipid catabolic processes in host plant cells, and may have increased nematode entry into plant tissues [14]. Similarly, the maximum binding energy of (2S)-2-amino-3-phenylpropanoic acid with G protein-coupled receptor kinase (−7.2 kcal/mol) could regulate cytological protein phosphorylation, nematode parasitism, nematode reproduction, and suppress the host defence response [34]. Furthermore, the maximum binding energy (−6.0 kcal/mol) for -1,4-endoglucanase with 2,6,10,15,19,23-hexamethyltetracosane could promote nematode activity to degrade the b 1,4 linkage in cellulose polymer, and could have promoted entry into plant tissues [35]. Following that, the maximum binding energy of pectate lyase with 2,6,10,15,19,23-hexamethyltetracosane could promote nematode activity to cleave the 1,4 glycosidic bonds of polygalacturonic acid in plant tissues [13]. Docking interactions of nematicidal compounds with a b-tubulin protein from *Brugia malayi* were conducted, and reported albendazol sulfone as that of the ideal nematicidal (antifilarial) substance [36]. Common chokepoint reactions and enzymes in nematodes and prioritised drug targets suggest perhexiline as a nematicidal compound, considering its binding efficacy against Caenorhabditis elegans carnitine palmitoyl transferase 2 [37]. Thus, the current study is unique in that phospholipase, one of the most targeted virulent effector proteins of *M. graminicola*, had the highest binding affinity to rice ligand biomolecules. Furthermore, this study’s findings highlighted the possibility of harnessing the potential of virulent effector protein phospholipase, which promotes parasitism with rice plants. However, more research is needed to confirm the pathogenicity of virulent effector phospholipase B-like protein in the wet lab using qRT-PCR and RNA interference techniques in rice plants challenged with rice root-knot nematode *M. graminicola*.

## 5. Conclusions

The current study provides a thorough molecular and morphological description of the *M. graminicola* population found in Tamil Nadu, India, as well as phase contrast microscopy analyses of adult females, second-stage juveniles, and posterior cuticular patterns. Furthermore, this study integrates *M. graminicola*’s life span and provides detailed morphometric and molecular assessments of all Tamil Nadu populations. Further, the present study used an in silico approach to explain the multiple modes of action of *M. graminicola*’s virulent proteins against various ligand biomolecules of rice targets. The virulent phospholipase B-like protein has a higher affinity for ligand biomolecules, which favours plant parasitism in rice.

## Figures and Tables

**Figure 1 biology-12-00987-f001:**
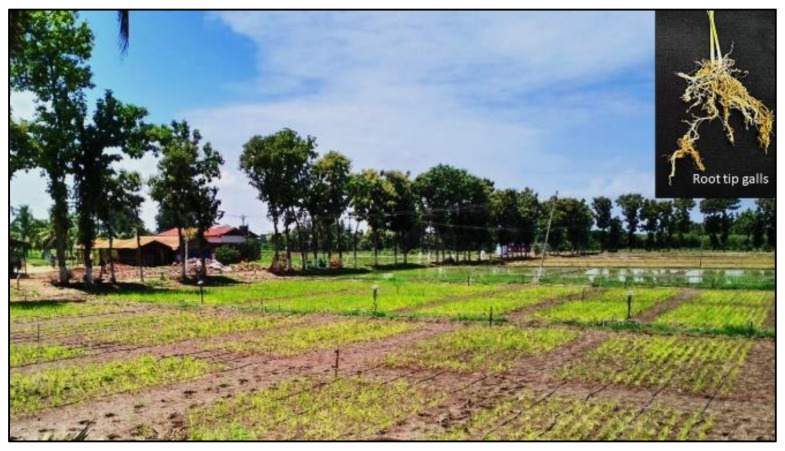
Rice root-knot nematode *Meloidogyne graminicola* infestation (a drip-irrigated rice field infested with root galling symptoms).

**Figure 2 biology-12-00987-f002:**
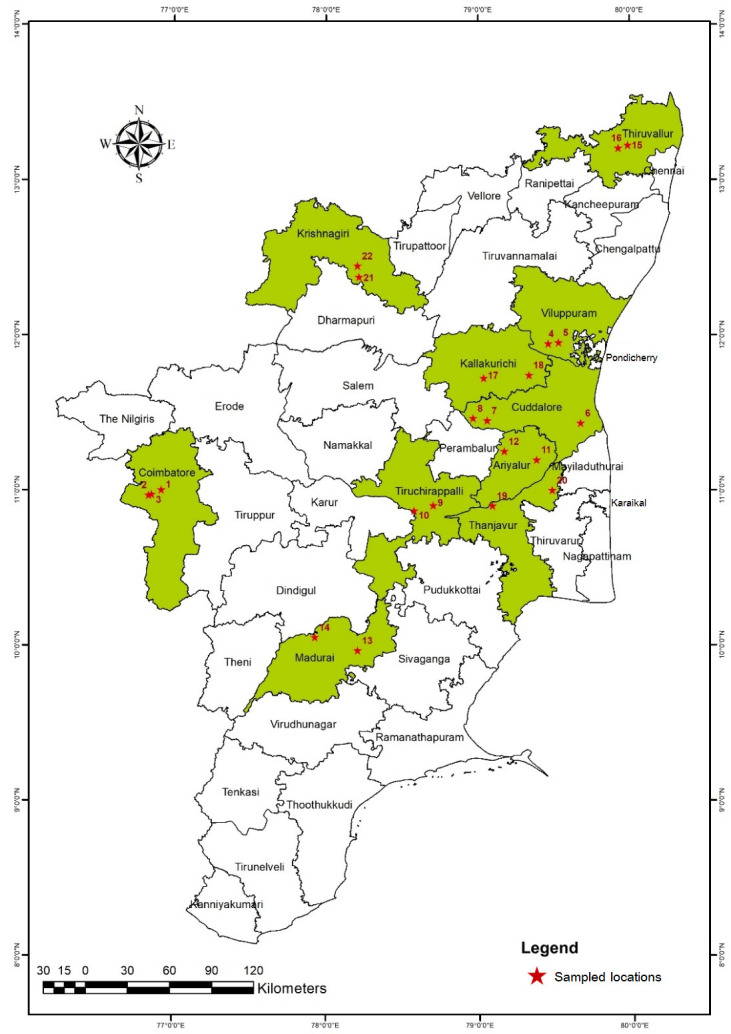
Surveyed regions of Tamil Nadu for the incidence of the rice root-knot nematode *Meloidogyne graminicola*. (red star sampled locations (Table 1)).

**Figure 3 biology-12-00987-f003:**
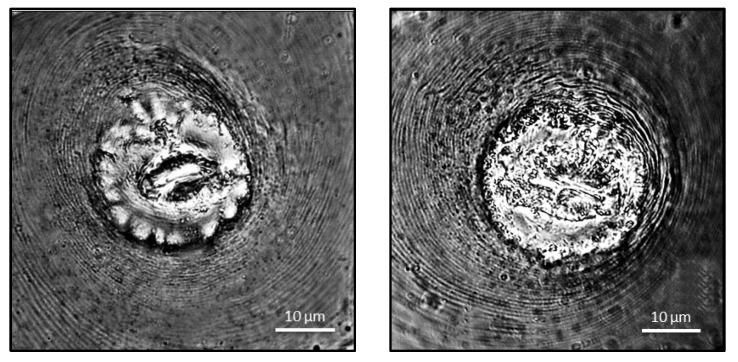
Perineal pattern of *Meloidogyne graminicola*.

**Figure 4 biology-12-00987-f004:**
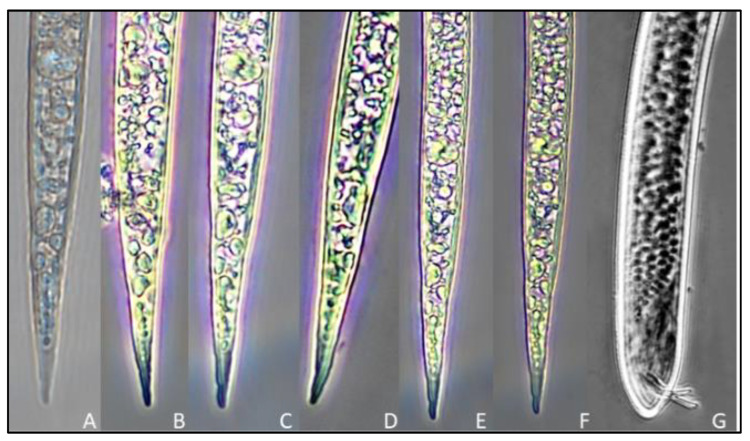
Tail regions of second-stage juveniles and male rice root-knot nematode *Meloidogyne graminicola* ((**A**–**F**)—tail images of second-stage juveniles, (**G**)—tail image of male nematode).

**Figure 5 biology-12-00987-f005:**
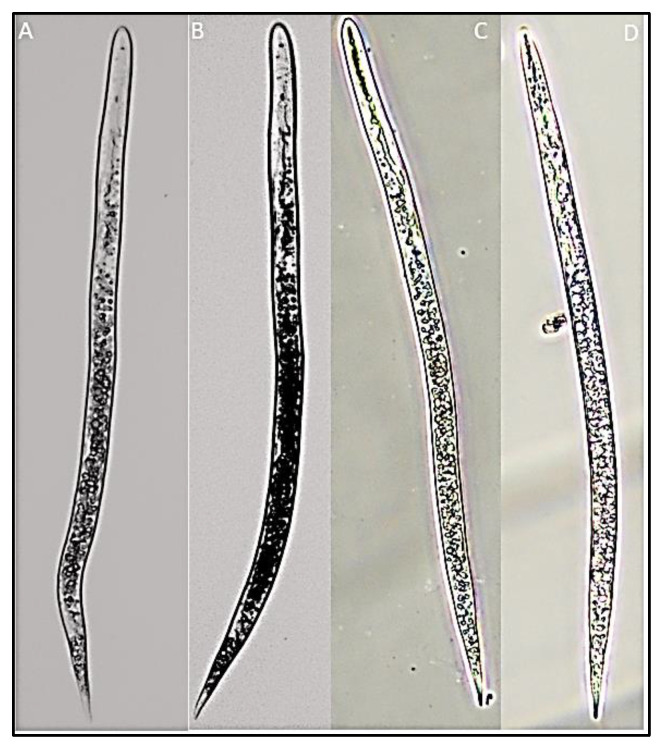
Second-stage juveniles of rice root-knot nematode *Meloidogyne graminicola* ((**A**–**D**)—second-stage juveniles).

**Figure 6 biology-12-00987-f006:**
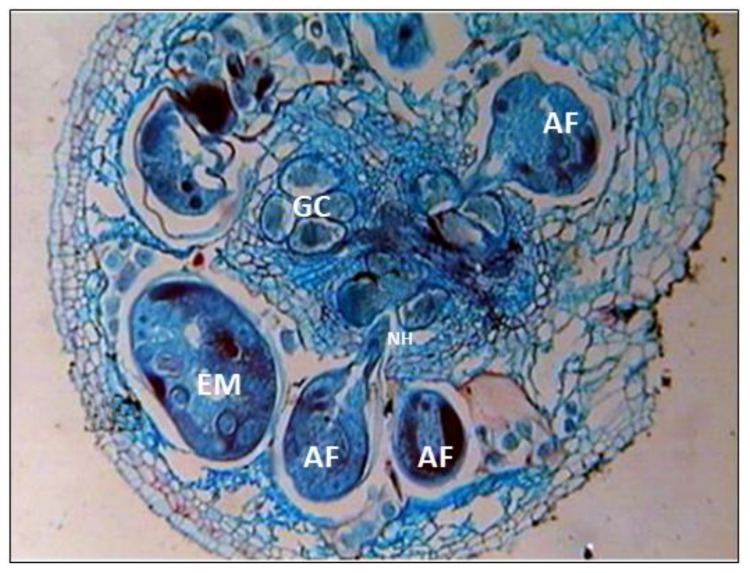
Histopathology of *M. graminicola* infested rice roots (AF—adult female; GC—giant cells; EM—egg mass with gelatinous matrix; NH—nematode head region).

**Figure 7 biology-12-00987-f007:**
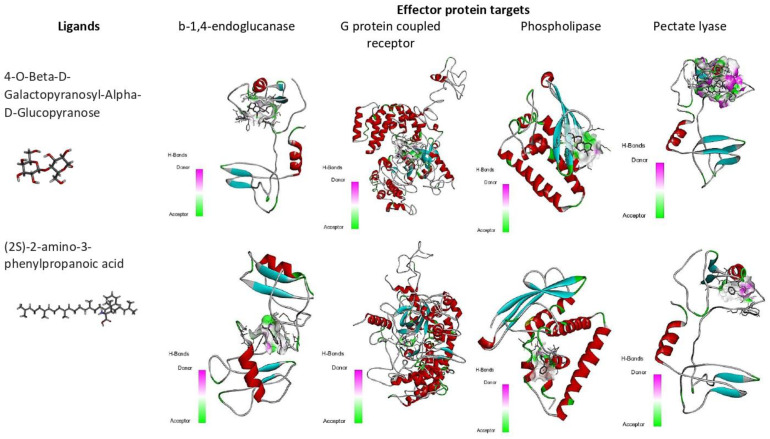
Binding affinity (kcal/mol) of small molecules, Beta-D-Galacturonic Acid, and (2S)-2-amino-3-phenylpropanoic acid on different *M. graminicola* effector proteins.

**Figure 8 biology-12-00987-f008:**
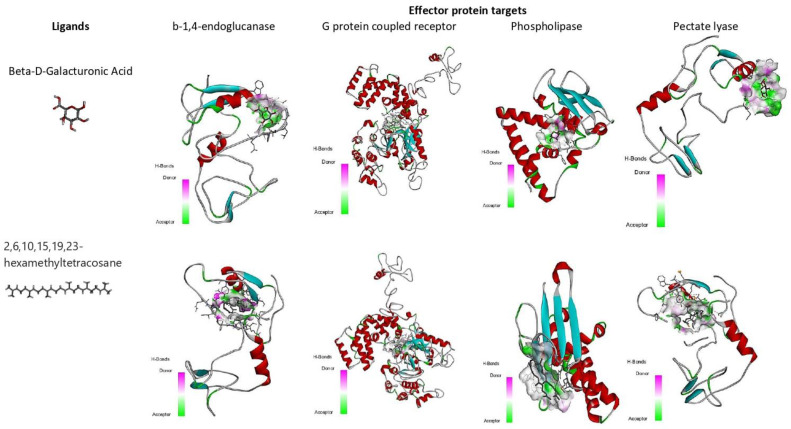
Binding affinity (kcal/mol) of small molecules, 2,6,10,15,19,23-hexamethyltetracosane, and 4-O-Beta-D-Galactopyranosyl-Alpha-D-Glucopyranose on different *M. graminicola* effector proteins.

**Figure 9 biology-12-00987-f009:**
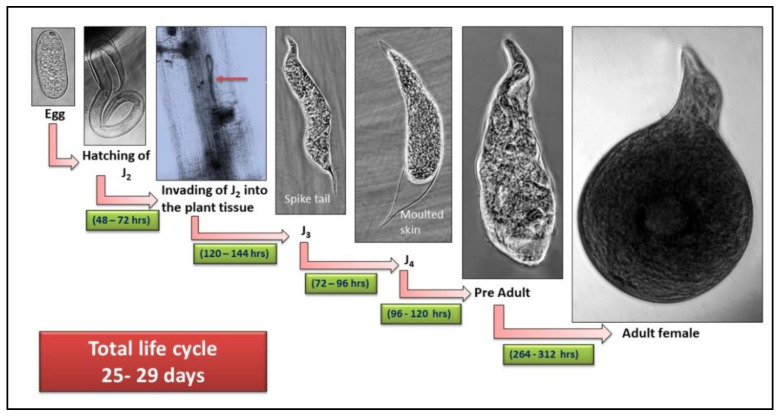
Life stages of rice root-knot nematode *Meloidogyne graminicola*.

**Table 1 biology-12-00987-t001:** Survey for the occurrence of *Meloidogyne graminicola* in rice fields of Tamil Nadu, India.

Location No.	Latitude	Longitude	Location Name	Prevalence	
				2021	2022
1	10.969365	76.843007	Thondamuthur	+	+
2	10.975203	76.862122	Madhampatti	+	+
3	11.003463	76.609100	Wet land, TNAU	+	+
4	11.944221	79.456656	Vedampathy	−	−
5	11.952058	79.523946	Poyyapakkam	−	−
6	11.432035	79.665464	Puvangiri	−	−
7	11.450248	79.054706	Alambadi	−	−
8	11.465288	78.961876	Pulikarambalur	−	−
9	10.903344	78.701033	Mannachanallur	−	−
10	10.870285	78.577262	Andhanallur	−	−
11	11.196512	79.377985	Pappankulam	−	−
12	11.251698	79.165723	Sendhurai	+	+
13	9.967661	78.207193	AC and RI, TNAU, Madurai	−	−
14	10.053014	77.929262	Karupatti	−	−
15	13.22259	79.982963	Vilampakkam	+	+
16	13.204888	79.920946	Seeyenjevi	−	−
17	11.72379	79.031681	Kurur	−	−
18	11.742295	79.328833	Padur	−	−
19	10.899854	79.087559	Perumpuliyur	−	−
20	10.999341	79.479342	TRRI, TNAU, Aduthurai	−	−
21	12.378174	78.216713	Paiyur	+	+
22	12.449002	78.207164	Kalvehalli	−	−

(‘+’—Presence of *M. graminicola*; ‘−’—absence of *M. graminicola*).

**Table 2 biology-12-00987-t002:** Occurrence of the rice root-knot nematode *Meloidogyne graminicola* in rice fields of Tamil Nadu, India.

Nematode Survey—2021				
Location	Galls/5 g Root Sample	Second-Stage Juveniles/200 cc Soil	No. of Females/ 5 g Root Sample	Gall Index
Thondamuthur	38.0 ± 1.36	567.2 ± 6.65	36.0 ± 1.36	4
Madhampatti	48.0 ± 2.14	861.7 ± 13.88	40.6 ± 1.85	4
Wetland, TNAU	52.0 ± 2.28	975.4 ± 10.38	46.4 ± 1.60	4
Sendhurai	18.2 ± 1.26	267.4 ± 6.34	20.5 ± 1.02	3
Vilampakkam	15.5 ± 1.62	321.6 ± 4.80	16.2 ± 0.98	3
Paiyur	38.2 ± 1.94	604.5 ± 5.82	33.4 ± 1.17	4
**Nematode Survey—2022**				
**Location**	**Galls/5 g Root Sample**	**Second-Stage Juveniles/200 cc Soil**	**No. of Females/** **5 g Root Sample**	**Gall Index**
Thondamuthur	38.0 ± 1.41	867.2 ± 12.92	36.0 ± 1.67	4
Madhampatti	44.8 ± 1.33	884.6 ± 13.31	38.2 ± 1.92	4
Wetland, TNAU	57.8 ± 1.72	865.1 ± 11.43	49.8 ± 1.36	4
Sendhurai	8.4 ±1.59	87.5 ± 4.41	5.6 ± 0.78	2
Vilampakkam	9.8 ± 1.17	324.4 ± 7.73	9.2 ± 1.34	2
Paiyur	42.5 ± 1.85	654.4 ± 7.06	38.2 ± 1.87	4

Data represented as mean ± standard deviation value of five subsamples.

**Table 3 biology-12-00987-t003:** Morphometric variations of females of *M. graminicola*.

Morphometric Parameters	Current Study	Golden and Birchfield (1965) (Original Description)	Zhao et al. (2001) [27]	Liu et al. (2011) [28]	Feng et al. (2017) [29]	Song et al. (2017) [30]	Zhong-ling et al. (2018) [7]
Origin	Tamil Nadu, India	Baton Rouge, Louisiana, USA	Hainan	Fujian	Jiangsu	Hunan	Zhejiang, China
Host	*Oryza sativa*	*Allium fistulosum*	*O. sativa*	*O. sativa*	*O. sativa*	*O. sativa*	*O. sativa*
n	15	20	20	20	20	20	7
L	564 (474–795)	573 (445–765)	-	613 (500–700)	585.2	619.1 (478.7–743.7)	598.9 (499.1–818.7)
a	1.49 (1.2–1.6)	1.37 (1.2–1.8)	-	1.45 (1.22–1.97)	1.4	-	1.8 (1.6–2.0)
Body Width	378 (383–480)	419 (275–520)	-	334 (245–485)	438.7	346.3 (242.6–525.8)	354.5 (277.3–455.5)
Stylet	10.8 (10.2–11.4)	11.08 (10.6–11.2)	12.1 (10.8–14.0)	10.9 (7.5–15)	11.3	12.8 (10.5–14.8)	10.2 (8.1–12.6)
DOGO	3.9 (3.6–5.9)	3.2 (2.8–3.9)	4.3 (3.7–4.7)	4.04 (2.5–5.0)	3.9	4.1 (3.5–5.1)	3.7 (2.9–4.9)
Vulval Slit Length	19.9 15.7–26.4	-	22.0 (18.8–27.5)	23.8 (15.0–29.8)	23.0	24.9 (16.6–30.4)	20.7 (17.3–25.5)

n = number of specimens on which measurements are based; L = body length; a = total body length/maximum body diameter; DOGO = dorsal oesophagus gland orifice; all measurements are in µm and in the form: means (range). -, no data.

**Table 4 biology-12-00987-t004:** Morphometric variations of males of *M. graminicola*.

Morphometric Parameters	Current Study	Golden and Birchfield (1965) (Original Description)	Zhao et al. (2001) [27]	Liu et al. (2011) [28]	Feng et al. (2017) [29]	Song et al. (2017) [30]	Zhong-ling et al. (2018) [7]
Origin	Tamil Nadu, India	Baton Rouge, Louisiana, USA	Hainan	Fujian	Jiangsu	Hunan	Zhejiang, China
Host	*Oryza sativa*	*Allium fistulosum*	*O. sativa*	*O. sativa*	*O. sativa*	*O. sativa*	*O. sativa*
n	12	20	5	20	15	20	17
L	1378 (1096.1–1731.4)	1222 (1020–1428)	1295 (1000–1515)	1421 (1150–1650)	1302.9	1475 (1246.1–1832.3)	1270 (1043.4–1553.4)
a	39.8 (36.2–45.8)	-	-	40.4 (35.0–48.3)	39.6	-	41.5 (38.48–53.3)
Body Width	34.6 (26.5–34.7)	29.8 (24–34.7)	-	35.4 (28.0–40.0)	32.6	39.0 (30.6–48.4)	25.9 (21.3–31.5)
Stylet	17.8 (17.2–19.1)	16.8 (16.2–17.3)	16.4 (16.0–17.2)	16.7 (15.0– 20.0)	17.3	19.3 (17.9–20.6)	17.2 (15.2–18.9)
DOGO	3.3 (2.8–4.0)	3.3 (2.8–4.0)	3.3 (3.0–3.8)	3.33 (2.9–4.0)	3.3	3.33 (2.9–4.6)	3.6 (3.2–4.0)
Tail Length	12.1 (9.8–15.1)	11.1 (6.1–15.1)	-	50.2 (40.0–58.1)	10.8	11.3 (8.4–16.6)	9.2 (8.1–10.3)
Spicule Length	25.6 (19.8–30.1)	28.1 (27.4–29.1)	-	23.9 (17.5–32.5)	27.2	30.7 (27.2–36.1)	21.1 (20.1–21.9)

n = number of specimens on which measurements are based; L = body length; a = total body length/maximum body diameter; DOGO = dorsal oesophagus gland orifice; all measurements are in µm and in the form: means (range). -, no data.

**Table 5 biology-12-00987-t005:** Morphometric variations of second-stage juveniles (J2) of *M. graminicola*.

Morphometric Parameters	Current Study	Golden and Birchfield (1965) (Original Description)	Zhao et al. (2001) [27]	Liu et al. (2011) [28]	Feng et al. (2017) [29]	Song et al. (2017) [30]	Zhong-Ling et al. (2018) [7]
Origin	Tamil Nadu, India	Baton Rouge, Louisiana, USA	Hainan	Fujian	Jiangsu	Hunan	Zhejiang, China
Host	*Oryza sativa*	*Allium fistulosum*	*O. sativa*	*O. sativa*	*O. sativa*	*O. sativa*	*O. sativa*
n	15	20	25	20	20	20	20
L	461.66 (412–484)	441 (415–484)	456.4 (410–510)	433 (376–480)	447.4	483 (427–514.9)	456.7 (402.7–509.0)
a	23.8 (22.3–27.8)	24.8 (27.3–22.3)	-	29.9 (25.4–34.3)	27.6	-	28.6 (23.0–32.9)
b	4.2 (3.2–4.5)	3.2 (2.9–4.0)	-	-	-	-	-
c	5.92 (5.2–6.9)	6.2 (5.5–6.7)	-	6.14 (5.35–7.26)	-	-	6.2 (5.5–6.7)
Body Width	16.9 (13.8–20.9)	-	-	15.2 (13.8–17.5)	16.3	17.5 (15.5–20.0)	16.1.5 (12.9–19.1)
Stylet	10.9 (11.2–12.6)	11.38 (11.2–12.3)	-	13.7 (13.0–15.0)	13.2	12.2 (11.4–12.8)	12.1 (10.6–13.1)
DOGO	3.2 (2.8–3.7)	2.83 (2.8–3.3)	-	4.7 (3.0–5.0)	4.0	-	2.6 (2.1–2.8)
Tail	71.4 (67.5–73.2)	70.9 (67.0–76.0)	72.9 (60.0–85.0)	70.2 (65.0–77.5)	71.6	-	70.2 (61.2–79.8)

n = number of specimens on which measurements are based; L = body length; a = total body length/maximum body diameter; b = total body length/distance from anterior to DOGO; c = total body length/tail length; DOGO = dorsal oesophagus gland orifice; all measurements are in µm and in the form: means (range). -, no data.

**Table 6 biology-12-00987-t006:** Binding affinity of Beta-D-Galacturonic Acid, 2,6,10,15,19,23-hexamethyltetracosane, (2S)-2-amino-3-phenylpropanoic acid, and its virulent targets.

Targets	Binding Affinity (kcal/mol) of Small Molecules on Different Targets	H-Bonds Formed
	Beta-D-Galacturonic Acid	2,6,10,15,19,23-hexamethyltetracosane	(2S)-2-amino-3-phenylpropanoic acid	4-O-Beta-D-Galactopyranosyl-Alpha-D-Glucopyranose	Beta-D-Galacturonic Acid	2,6,10,15, 19,23-hexamethyltetracosane	(2S)-2-amino-3-phenylpropanoic acid	4-O-Beta-D-Galactopyranosyl-Alpha-D-Glucopyranose
	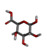	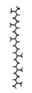	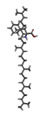	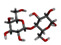	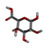	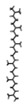	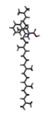	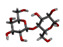
β-1,4-endoglucanase	−5.1	−6.0	−5.2	−5.2	THR 354, ALA 356, THR 361	THR 332, THR 334, THR 335, THR 392	GLU 156, LYS 283	ILE 75, TYR 149
G protein-coupled receptor kinase	−5.8	−7.3	−7.2	−6.7	LYS 237, LYS 271	LYS 277	LYS 237, THR 273	ASP 279, ALA 305, SER 318, ASP 319
Phospholipase	−5.4	−7.7	−5.5	−5.5	GLU 154, LEU 162	TYR133, GLU131	GLN 313	ARG 72
Pectate lyase	−6.8	−7.5	−6.9	−6.4	PHE 62, ASP 60, ALA 58,GLY 55	TYR133, GLU131	PRO 52	VAL 53

The two types of hydrogen bonds formed between docked complexes are those with the backbone and side-chain of amino acid residues. Other types of contacts were visible in the docked complex, including hydrophobic interactions, van der Waals, pi–pi, alkyl, and pi–alkyl.

## Data Availability

Not applicable.

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
