# Peer review of "An Insight into Occurrence, Biology, and Pathogenesis of Rice Root-Knot Nematode Meloidogyne graminicola"

_biology, 2023, doi:10.3390/biology12070987_

Round 1

Reviewer 1 Report

Please correct the minors and complete some information in MM section.

L98: Are root galls present on the crops' above-ground?

103: “three samples were taken from each plant”, what kind of sample was taken from each plant?

108: “ Rice root-knot nematode, Meloidogyne graminicola infestation”: how did the author confirm these phenomena caused by Mg? whereas “morphological characterization of Meloidogyne spp.” (L112)

127:  “from guava root gall”: is this true? And why is guava?

138: Cited the origin of primers, please!

160-163: “from a paddy field that had been infected with M. graminicola and contained one second stage juvenile (J2) per gram of soil … so that life cycle research could be conducted” : please give more details on how carrying and references.

168-169: “Root samples from rice plants that had been severely infested with root galls caused by the root-knot nematode M. graminicola were used in this study”: how did the authors confirm this?

Table 4: please give the SD value after the mean.

Figure 6: The image is very dark, it is not possible to observe the details as described in the results.

L 311: insilico à in silico

Author Response

Response to Reviewer Comments

L98: Are root galls present on the crops' above-ground?

Response L98: No, root galls were present in the below-ground root regions.

L103: “three samples were taken from each plant”, what kind of sample was taken from each plant?

Response L103: Root samples weighing 5 g and 200 cc of soil were taken from Meloidogyne graminicola-infested plants and its rhizosphere region respectively.

108: “ Rice root-knot nematode, Meloidogyne graminicola infestation”: how did the author confirm these phenomena caused by Mg? whereas “morphological characterization of Meloidogyne spp.” (L112)

Response 108: In contrast to normal plants, rice root-knot nematode-infested plants had root gall signs. Dissection of those galls revealed mature females that were white in colour, confirming that they were Meloidogyne spp. Further, the Meloidogyne graminicola identity was confirmed by Morphometrics [(Body length, Body width, Stylet Length, DOGO), Posterior Cuticular Pattern (oblong shaped, with fine striae, dorsal semicircular arches, and occasionally very few lines converged at each end of the vulva (Tian et al. 2018)] and Molecular characterization (NCBI GenBank Accession No - OP712502, OP714360 and OP714470).

L127:  “from guava root gall”: is this true? And why is guava?

Response L127: No, that was a mistaken typing typo that has been altered in the manuscript to "rice."

L138: Cited the origin of primers, please!

Response L138: Primers references were cited in accordance with the reviewer's remarks. Further the reference numbers were corrected in the manuscript accordingly.

L160-163: “from a paddy field that had been infected with M. graminicola and contained one second stage juvenile (J2) per gram of soil … so that life cycle research could be conducted” : please give more details on how carrying and references.

Response L160-163: Meloidogyne graminicola's life cycle study was conducted using soil that was taken from a nematode-infested field and included one second stage juvenile per gramme of soil, which is in compliance with the economic threshold level for plant parasitic nematodes (Dabur et al., 2004) and the reference was cited in the manuscript.

L168-169: “Root samples from rice plants that had been severely infested with root galls caused by the root-knot nematode M. graminicola were used in this study”: how did the authors confirm this?

Response L168-169: Meloidogyne graminicola used in this whole experiment was preliminarly confirmed by morphotry and molcular technique. Then, the Meloidogyne graminicola pure culture was established from a single egg mass dissected from infested root galls of M. graminicola. This pure culture was used for the whole experiment.

Table 4: please give the SD value after the mean.

Response Table 4: The manuscript has been revised in accordance with the reviewer's suggestions.

Figure 6: The image is very dark, it is not possible to observe the details as described in the results.

Response Figure 6: The clear image was replaced now in the manuscript as per the reviewer comment.

L 311: insilico à in silico

Response L 311: The manuscript has been revised in accordance with the reviewer's suggestions.

Reviewer 2 Report

I think you must review the order of citation of figure 1 and figure 2. Usually the figures are inserted in order of citation. Figure 1 should be the first cited and Figure 2 the second to be cited.

I believe you have done substantial work, which can be published.

Author Response

Response to Reviewer Comments

L99: Figure 1 should be the first cited and Figure 2 the second to be cited.

Response L 99: The manuscript has been revised in accordance with the reviewer's suggestions.

Reviewer 3 Report

The paper aims to provide insight into various aspects of biology and distribution of the rice root-knot nematode in the southeastern region of Tamil Nadu, India. However, the various sections of the work are plenty of widely known, often redundant information, with relatively little novelty.

The paper appears incoherently structured and seems not to fulfill the expectations highlighted in the title.

The Simple Summary includes a list of obvious sentences, with no mention or remarks of any of the issues treated in the paper.

The morphological studies are quite superficially conceptualized and conducted; data on molecular identification are not presented, unless in a short sentence in the Discussion section. The histopathology section does not add any insight to what is already known from the literature. Even molecular docking studies, carried out by an in silico approach, mostly rely on speculation on the mode of pathogenicity of a few effector proteins, going not deep to unravel the mechanisms underlying the interactions between the nematode and the target sites.

The manuscript contains several typing errors (especially in section 3.2, with several full stops not followed by capital letters). Several references in the text are improperly cited (i.e. citations [9] and [23] in paragraphs 2.6.4 and 2.6.5, or [5] in the Discussion section, at line 394) and do not correspond to the wanted cited article. Meloidogyne graminicola and other Latin species names are not italicized several times throughout the text as well as in the References, whereas they should be not in italic within the headings of paragraphs (i.e. 2.4, 2.5, etc.). A few sentences are unclear (i.e. lines 19-21 and 77-78), and some others reveal the use of improper terminology or need some editing.

The Introduction is extremely long and needs to be shortened. The sentence of lines 55-57 should be deleted.

In M&M, the volume of 200 cc to make a composite sample appears rather small. The procedure of nematode extraction mentions two methods but is unclear which one of the two was followed by the authors. In section 2.3 is stated ‘six females were detached from guava root…’ Do the authors mean ‘rice’ roots? Were they not extracted from M. graminicola-affected rice plants in the six Tamil Nadu sampling sites?

In my opinion, rather than showing Table 3, the authors could refer to some already published protocols, or the manufacturer’s conditions, and explain the procedures they followed, avoiding stating: ‘Gel ….can be performed’, or ‘The gel can be visualized…’ (lines 143-145).

Results concerning morphology are poorly presented. The authors show in Tables 5 (reported twice for two different tables!) and Table 6 measurements of adult and juvenile stages, but they do not mention where the nematode population they measured came from, among the six sampled locations infested by M. graminicola. They also show perineal patterns of just two female specimens, stating they “…are somewhat different” from standard M. graminicola patterns (citation [23] in this sentence is not pertinent), and overlap with those of other two RKN species attacking rice, thus devaluing the results of their morphological analysis. Descriptions of female (“Figure 3” in the text!), males and juveniles report unclear sentences (such as in lines 266-267 and 276-277) and quite ‘unusual’ terms (i.e. ‘lateral area’ in place of lateral fields, or “…four latitudinally incised fields”, to mean lateral fields bearing four longitudinal lines; DOGO = dorsal oesophageal gland orifice/or outlet, without “Pharyngeal”). Moreover, stylet knobs of J2 are usually never “prominent”, and Figure 4G does not exist. In Table 5 (with morphometrics of males), values reported for tail length in the column of measurements by Liu et al. are abnormally long.

The Discussion paragraph is long and contains sentences that should be moved to Results, such as: the sequence size of ITS and other molecular markers (line 397), as well as all observations on the life stage and mode of parasitism (lines 409 to 427). Newly submitted accession numbers could be better reported in a Table apart, rather than within the text. Sentence in lines 431-434 can be deleted. Sentences in lines 444-450 should be rephrased and shortened. Sentences in lines 466-471 are unclear and should be rephrased for clarity.

Finally, in the Conclusions section, it is stated that a ‘phase contrast microscopy analyses’ of females, J2 and female perineal patterns are provided, although no mention of this is made in Materials and Methods.

Minor issues concern:

Page1, line 31: M. graminicola is misspelled;

Page 5, l. 115-116: The authors should avoid describing the composition of fixative FA 4:1; they could rather refer to Hooper’s (1986) or any other manual of laboratory techniques for nematology;

Page 5, l. 124-125: how can a publication dated 1880 be considered to contain “revised criteria for species diagnosis”?

P. 6, l. 153: Neighbor-Join should be Neighbor-joining;

P. 8, Table 4: change ‘s’ (no capital) of Nematode Survey – 2022;

P. 8, l. 255: I disagree with the acronym PCP; the authors should better refer to ‘perineal patterns’; moreover, I would change the statement “…morphological diagnosis based on…has been the accepted method…” to “is one of the traditional (or commonly used) methods for…etc.”;

P. 9, l. 275: “A hemizonid with three annules…” should be: “Hemizonid, corresponding to three body annuli in length, just anterior to excretory pore”;

P. 10 Table 5: N (first column) should not be in capital letter;

P. 13, l. 311: 3.4. ‘insilico’ should be In silico;

P. 16, Figure 10.b: the figure is little informative and could be omitted.

Author Response

Response to Reviewer Comments

Page1, line 31: M. graminicola is misspelled 

Response Page1, line 31: The manuscript has been revised in accordance with the reviewer's suggestions.

Page 5, l. 115-116: The authors should avoid describing the composition of fixative FA 4:1; they could rather refer to Hooper’s (1986) or any other manual of laboratory techniques for nematology

Response Page 5, l. 115-116: The manuscript has been revised in accordance with the reviewer's suggestions and the reference (Hoopers (1986) was cited.

Page 5, l. 124-125: how can a publication dated 1880 be considered to contain “revised criteria for species diagnosis?

Response Page 5, l. 124-125: The manuscript has been updated as “revised criteria for species diagnosis” to “De Man criteria (1880) for species diagnosis”.

P. 6, l. 153: Neighbor-Join should be Neighbor-joining

Response P. 6, l. 153: The manuscript has been revised in accordance with the reviewer's suggestions as Neighbor-Join to “Neighbor-joining”

P. 8, Table 4: change ‘s’ (no capital) of Nematode Survey – 2022!

Response 8, Table 4: The manuscript has been updated with mentioned correction in Table. 4

P. 8, l. 255: I disagree with the acronym PCP; the authors should better refer to ‘perineal patterns’; moreover, I would change the statement “…morphological diagnosis based on…has been the accepted method…” to “is one of the traditional (or commonly used) methods for…etc.”;

Response P. 8, l. 255: The acronym PCP was changed as “perineal pattern” in the manuscript and mentioned corrections were made in the manuscript.

9, l. 275: “A hemizonid with three annules…” should be: “Hemizonid, corresponding to three body annuli in length, just anterior to excretory pore

Response P. 9, l. 275: The manuscript has been revised in accordance with the reviewer's suggestions.

P.10 Table 5: N (first column) should not be in capital letter

Response P. 10 Table 5: The manuscript has been revised in accordance with the reviewer's suggestions in Table 5.

P.13, l. 311: 3.4. insilico’ should be In silico.

Response P. 13, l. 311: 3.4: The term ‘insilico’ was changed as “In silico” in the manuscript.

P. 16, Figure 10.b: the figure is little informative and could be omitted.

Response P. 16, Figure 10.b: Figure 10 b removed from the manuscript in accordance with the reviewer's suggestions.

Round 2

Reviewer 3 Report

The authors improved the all part concerning the morphological description of the species, and replied to the ‘minor’ issues, although they substantially ignored several other suggestions.

A few things still need to be addressed:

·         In the Abstract, Oryza sativa should be italicized, as well as Meloidogyne in Introduction (line 72);

·         Page 3, line 104: “Figure 2” Table 1 are cited. However, Figure 2 does not exist (!), and ‘Table 1’ should rather be Table 2?

·         Page 4, l 113: ‘glass rod’ should be plural (glass rods); same page, line 116, separate ‘microscopewas’ and on line 117, delete the word ‘photo’ before microphotographs;

·         Page 4, l. 119: De Man’s criteria are too old for a modern description and comparison of the species. Please refer to Siddiqi’s book (2000) or more updated references (i.e. Ye and Hunt, 2021 in  Techniques for Work with Plant and Soil Nematodes, CABi);

·         P. 7, lines 258-59, change the sentence: ‘There is some cephalic frame work present, although it is not significant’ with ‘Cephalic framework weakly sclerotized’.

·         P.8, l. 261: ‘metacarpus’ should be metacorpus; on the same page, line 275 Figure 4 should be Figure 5;

·         P. 10, l. 294 and p. 11, l. 299: delete the word ‘Pharyngeal’;

·         Please check the values of male tail length in the population of Liu et al (2011) in Table 4: they are incongruent!

·         P.12, l. 324: what is Plate 9j?

·         In ‘Discussion’, line 417: delete ‘Dorsal Oesophagus gland Opening (DOGO) leaving just ‘DOGO’ (acronyms are meant to this purpose).

Author Response

Response to Reviewer Comments

In the Abstract, Oryza sativa should be italicized, as well as Meloidogyne in Introduction (line 72);

Response line 72: The manuscript has been revised in accordance with the reviewer's suggestions.

Page 3, line 104: “Figure 2” Table 1 are cited. However, Figure 2 does not exist (!), and ‘Table 1’ should rather be Table 2?

Response Page 3, line 104: Figure 2 was inserted and Table 2 was cited in accordance with reviewers suggestions.

 Page 4, l 113: ‘glass rod’ should be plural (glass rods); same page, line 116, separate ‘microscopewas’ and on line 117, delete the word ‘photo’ before microphotographs

Response Page 4, l 113: The manuscript has been updated as per the comments.

Page 4, l. 119: De Man’s criteria are too old for a modern description and comparison of the species. Please refer to Siddiqi’s book (2000) or more updated references (i.e. Ye and Hunt, 2021 in  Techniques for Work with Plant and Soil Nematodes, CABi);

Response P. 6, l. 153: The manuscript has been revised in accordance with the reviewer's suggestions as De Man’s criteria was replaced with Ye and Hunt, 2021 and cited in the references

P. 7, lines 258-59, change the sentence: ‘There is some cephalic frame work present, although it is not significant’ with ‘Cephalic framework weakly sclerotized’.

Response P. 7, lines 258-59: The manuscript has been updated with mentioned corrections.

P.8, l. 261: ‘metacarpus’ should be metacorpus; on the same page, line 275 Figure 4 should be Figure 5

Response P. 8, l. 261: The manuscript has been revised in accordance with the reviewer's suggestions.

P. 10, l. 294 and p. 11, l. 299: delete the word ‘Pharyngeal’;

Response P. 10, l. 294 and p. 11, l. 299: the word ‘Pharyngeal’ removed from the l. 294 and l. 299

 Please check the values of male tail length in the population of Liu et al (2011) in Table 4: they are incongruent!

The values of male tail length was verified with Liu et al (2011) with the same value mentioned in the manuscript . (DOI:10.3969/j.issn.1001-7216.2011.04.012)

 P.12, l. 324: what is Plate 9j?

Response P.12, l. 324: The typo error happened unknowingly and it was replced as Figure 6.

In ‘Discussion’, line 417: delete ‘Dorsal Oesophagus gland Opening (DOGO) leaving just ‘DOGO’ (acronyms are meant to this purpose).

Response line 417: The manuscript has been revised in accordance with the reviewer's suggestions